# CXCL12 Neutralizing Antibody Promotes Hair Growth in Androgenic Alopecia and Alopecia Areata

**DOI:** 10.3390/ijms25031705

**Published:** 2024-01-30

**Authors:** Mei Zheng, Min-Ho Kim, Sang-Gyu Park, Won-Serk Kim, Sang-Ho Oh, Jong-Hyuk Sung

**Affiliations:** 1Epi Biotech Co., Ltd., Incheon 21983, Republic of Korea; zhengmei0223@hanmail.net (M.Z.); kmh0212@epibiotech.com (M.-H.K.); 2College of Pharmacy, Ajou University, Suwon 16499, Republic of Korea; sgpark@ajou.ac.kr; 3Department of Dermatology, School of Medicine, Sungkyunkwan University, Kangbuk Samsung Hospital, Seoul 03181, Republic of Korea; susini@naver.com; 4Department of Dermatology and Cutaneous Biology Research Institute, Severance Hospital, Yonsei University College of Medicine, Seoul 03722, Republic of Korea; oddung93@yuhs.ac

**Keywords:** CXCL12, androgenic alopecia, alopecia areata, neutralizing antibody, dermal fibroblasts

## Abstract

We had previously investigated the expression and functional role of C-X-C Motif Chemokine Ligand 12 (CXCL12) during the hair cycle progression. CXCL12 was highly expressed in stromal cells such as dermal fibroblasts (DFs) and inhibition of CXCL12 increased hair growth. Therefore, we further investigated whether a CXCL12 neutralizing antibody (αCXCL12) is effective for androgenic alopecia (AGA) and alopecia areata (AA) and studied the underlying molecular mechanism for treating these diseases. In the AGA model, CXCL12 is highly expressed in DFs. Subcutaneous (s.c.) injection of αCXCL12 significantly induced hair growth in AGA mice, and treatment with αCXCL12 attenuated the androgen-induced hair damage in hair organ culture. Androgens increased the secretion of CXCL12 from DFs through the androgen receptor (AR). Secreted CXCL12 from DFs increased the expression of the AR and C-X-C Motif Chemokine Receptor 4 (CXCR4) in dermal papilla cells (DPCs), which induced hair loss in AGA. Likewise, CXCL12 expression is increased in AA mice, while s.c. injection of αCXCL12 significantly inhibited hair loss in AA mice and reduced the number of CD8^+^, MHC-I^+^, and MHC-II^+^ cells in the skin. In addition, injection of αCXCL12 also prevented the onset of AA and reduced the number of CD8^+^ cells. Interferon-γ (IFNγ) treatment increased the secretion of CXCL12 from DFs through the signal transducer and activator of transcription 3 (STAT3) pathway, and αCXCL12 treatment protected the hair follicle from IFNγ in hair organ culture. Collectively, these results indicate that CXCL12 is involved in the progression of AGA and AA and antibody therapy for CXCL12 is promising for hair loss treatment.

## 1. Introduction

Androgenic alopecia (AGA) is the most prevalent form of hair loss and is characterized by progressive hair thinning in a specific pattern. This condition is attributed to the influence of androgens, particularly dihydrotestosterone (DHT), on susceptible hair follicles (HFs). Recent research has unveiled intriguing insights into the potential role of CXCL12 as a therapeutic target for AGA. Originally, CXCL12 was thought a chemokine with multifaceted roles in cellular migration, tissue homeostasis, and wound healing; however, we first demonstrated its involvement in hair cycle regulation [1,2]. In addition, inflammatory mediators and immune mediators such as CXCL12 expression were significantly increased in the scalps of AGA patients [3]. Notably, CXCL12 appears to interact with the androgen signaling pathway, i.e., DHT and testosterone upregulated CXCL12 expression through the AR [4,5]. Moreover, CXCL12 induction occurs only in AR-positive breast cancer cell lines, via an androgen response element (ARE) upstream of the CXCL12 promoter [6]. Therefore, it is hypothesized in this study that androgens induced hair miniaturization via AR-induced CXCL12 and C-X-C Motif Chemokine Receptor 4 (CXCR4) in dermal papilla cells (DPCs) and/or dermal fibroblasts (DFs). Furthermore, understanding the intricate interplay between CXCL12, androgen signaling, and hair follicle biology holds promise for developing innovative therapeutic interventions.

Alopecia areata (AA) is an autoimmune disorder characterized by non-scarring hair loss resulting from immune system attack on HFs. Despite its non-life-threatening nature, AA can have profound psychosocial effects due to its unpredictable course and its impact on self-image. Emerging research has shed light onto the potential therapeutic role of CXCL12 in addressing AA. For example, CXCL12 plays pivotal roles in immune regulation and is involved in autoimmune diseases such as psoriasis and lupus [7]. In addition, pro-inflammatory γδ1^+^ T-cell infiltrates are present in and around the hair bulbs of AA patients and lesional AA HFs show significantly higher expression of CXCL12 [8]. γδ T cells have stress-surveillance functions and induced HF immune privilege collapse, dystrophy, and premature catagen by secreting CXCL12 [9]. Of note, it had been reported that CXCL12 stimulated CD4 and CD8 T cells, which play a key role in AA progression [10,11]. By inhibiting CXCL12 signaling, it might be possible to mitigate the immune attack on HFs, potentially leading to reduced hair loss in AA patients.

In a previous study, we have demonstrated the expression and functional role of CXCL12 during hair cycle progression. CXCL12 was highly expressed in stromal cells such as DFs, slightly in outer root sheath (ORS) cells, and its level was elevated throughout the catagen and telogen phases of the hair cycle. Hair loss was induced by recombinant CXCL12 therapy, which delayed the telogen-to-anagen transition in an animal model. In contrast, inhibition of CXCL12 using siRNA and antibodies promoted hair growth. Hair cycle regulation by CXCL12 was mediated by CXCR4, and inhibition of CXCR4 also increased hair growth [2]. However, we did not demonstrate the hair growth promoting effects of CXCL12 in AGA and AA animal models in that study. Therefore, we further investigated in the present study whether the CXCL12 neutralizing antibody (αCXCL12) is effective for treating AGA and AA in animal models and studied the underlying molecular mechanisms for treating these diseases.

## 2. Results

### 2.1. Hair Growth Promoting Effects of CXCL12 Neutralizing Antibody in Testosterone-Induced AGA Models

We first examined the expression of CXCL12 and effect of αCXCL12 in testosterone propionate (TP)-induced AGA models. Previously, we found that CXCL12 is co-localized with PDGFRα, which serves as a marker for stromal cells such as DFs [2]. Seven-week-old C_3_H male mice were subcutaneously injected with TP for 3 weeks (0.5 mg/head/day) and a TP-induced AGA model was prepared (Figure 1A) [12]. Based on confocal images, we observed a high expression of CXCL12 in the DFs of mouse skin treated with TP (Figure 1B). Compared with control, TP treatment significantly delayed the hair growth. However, subcutaneous (s.c.) injection of αCXCL12 (5 μg or 20 μg twice a week) showed a significant increase in hair weight (Figure 1C, ### *p* < 0.001 vs. control, * *p* < 0.05, ** *p* < 0.01 vs. TP-treated, n = 7). Further investigation on the effect of αCXCL12 in TP-induced AGA was conducted using mouse vibrissae organ culture models. TP treatment significantly reduced the hair growth, whereas αCXCL12 treatment increased the length of mouse vibrissae (Figure 1D, ## *p* < 0.01 vs. control, * *p* < 0.05 vs. TP-treated, n = 8).

### 2.2. Hair-Growth-Promoting Effects of αCXCL12 in Dihydrotestosterone-Induced AGA Models

We further examined the expression and effect of CXCL12 in DHT-induced AGA models. Seven-week-old C_3_H male mice were subcutaneously injected with a single dose of DHT (10 mg/head) and a DHT-induced AGA model was prepared (Appendix A) [13]. Similarly, CXCL12 was highly expressed in the DFs of mouse skin treated with DHT (Appendix A). Compared with the control, DHT treatment significantly delayed the hair growth. However, s.c. injection of αCXCL12 (20 μg twice a week) led to a significant increase in hair weight (Appendix A). DHT treatment also reduced the hair growth in hair organ culture, and αCXCL12 was found to increase the length of mouse vibrissae in a dose-dependent manner (Appendix A).

### 2.3. Androgens Increased the Secretion of CXCL12 from DFs through the AR

As CXCL12 expression in the skin was increased in AGA animal models, we hypothesized that increased CXCL12 secretion by DFs plays a key role in the progression of AGA. Therefore, the effect of androgens (DHT and TP) on the expression and secretion of CXCL12 was evaluated in human DFs. DFs were incubated with varying concentrations of DHT or TP (1~100 nM), resulting in a significant increase in CXCL12 mRNA levels compared with control (Figure 2A, $ *p* < 0.05 vs. control). Both DHT and TP treatment increased the CXCL12 expression in a dose-dependent manner. In addition, the secreted protein levels were measured using enzyme-linked immunosorbent assay (ELISA) and increased in response to androgen treatment (Figure 2B, $ *p* < 0.05, $$ *p* < 0.01 vs. control). We further examined whether androgens induce CXCL12 expression via AR. As shown in Figure 2C, DHT and TP treatment resulted in the translocation of AR to the nucleus in DFs. To confirm the involvement of AR in CXCL12 expression, we carried out an AR knockout (AR-KO) in DFs using the CRISPR/Cas9 system. DFs cultured and transfected with CRISPR/Cas9 showed a decreased AR expression (Figure 2D, right). AR-KO significantly decreased CXCL12 secretion after androgen treatment (Figure 2D, ### *p* < 0.001 vs. control, ** *p* < 0.01, *** *p* < 0.001 vs. TP or DHT treatment). Collectively, our results suggest that androgens induce CXCL12 expression in DFs and AR is involved in CXCL12 expression.

### 2.4. The CXCL12 Secreted from DFs Mediated the Hair Miniaturization Induced by Androgens

It has been well reported that androgens directly interact with AR in DPCs to induce hair miniaturization [14]. However, we hypothesized that the CXCL12 secreted from DFs might be responsible for the hair miniaturization induced by androgens. Therefore, we first examined the influence of recombinant human CXCL12 (rCXCL12) on the expression of AR and CXCR4 in DPCs. Treatment with rCXCL12 (3, and 10 ng/mL) in human DPCs resulted in significant increases in AR and CXCR4 mRNA and protein levels compared with control (Figure 3A–C, $ *p* < 0.05, $$ *p* < 0.01 vs. Control). To investigate whether the CXCL12 secreted by DFs plays a role in hair loss, we stimulated the secretion of CXCL12 by DFs using DHT and TP. Subsequently, we cultured human scalp hair follicles with the conditioned medium of DFs (DFCM), DHT-treated DFs (DFCM^DHT^), and TP-treated DFs (DFCM^TP^). We found that DFCM^DHT^ and DFCM^TP^, which contained high CXCL12 levels, significantly decreased the length of human HFs (Figure 3D, $ *p* < 0.05 vs. DFCM, n = 10). These findings suggest that the CXCL12 secreted from DFs plays a role in hair miniaturization by androgens.

### 2.5. Increased Expression of CXCL12 in an AA Mouse Model

Skin-draining lymph node (SDLN) cells from AA-affected mice were isolated and cultured, then intradermally injected into the dorsal skin of mice, resulting in AA development. Approximately 60–80% of the mice that received cultured SDLN cells developed AA after 12 weeks (Figure 4A). AA mice showed a high expression of CD8^+^ cells in the skin (Appendix A). We also observed a higher level of CXCL12 expression in skin of AA mice than in the skin of normal C_3_H mice (Figure 4B).

### 2.6. CXCL12 Neutralization Reversed AA in C_3_H Mice

To investigate whether αCXCL12 could reverse AA in the model, αCXCL12 was subcutaneously administrated twice (20 μg) after AA induction (Figure 4A; 0~5 week). After 5 weeks, mice treated with αCXCL12 showed hair regrowth in the skin, whereas control mice experienced progressive and extensive hair loss (Figure 4C, *** *p* < 0.001 vs. control, n = 3). A high percentage of CD8^+^ cells was detected in the lesional skin of AA nice (Appendix A) and αCXCL12 treatment reduced the CD8^+^ cells in the skin (Figure 4D upper panel). αCXCL12 treatment also reduced the expression of the MHC-I and MHC-II proteins in AA skin (Figure 4D middle and lower panel). These findings suggest that αCXCL12 is effective for AA treatment through reducing the immune reaction.

### 2.7. CXCL12 Neutralization Prevented AA Onset in C_3_H Mice

We investigated whether αCXCL12 treatment also could prevent the onset of AA. C_3_H mice were treated with αCXCL12 every week following transplantation of SDLN cells (Figure 5A). In the control group, seven out of ten mice were exhibiting hair loss by the 13th week following SDLN transplantation. In contrast, only one out of nine mice displayed a slight degree of hair loss in the αCXCL12-treatment group (Figure 5B,C, Control (n = 10) or αCXCL12 treated (n = 9), $$ *p* < 0.01 vs. control). Neutralizing CXCL12 also significantly reduced the number of CD8^+^ cells in the skin (Figure 5D). We also examined the NKG2D^+^ CD8^+^ T cells by isolating SDLN cells from control and αCXCL12-treated mice. CXCL12 blockade decreased the frequency of NKG2D^+^ CD8^+^ T cells in the SDLN (Appendix A).

### 2.8. IFNγ Increased the Secretion of CXCL12 through the STAT3 Pathway

Previous studies have indicated that IFNγ plays a significant role in the pathogenesis of AA [15,16,17]. Therefore, we examined the effects of IFNγ on the CXCL12 mRNA level and secreted protein level in DFs. As shown in Figure 6A,B, both the mRNA and secreted protein levels of CXCL12 were enhanced by IFNγ treatment in a dose-dependent manner ($ *p* < 0.05, $$$ *p* < 0.001 vs. control). It has also been reported that IFNγ regulated cytokine and chemokine secretion via the JAK/STAT signaling pathway; therefore, we further examined whether IFNγ induced CXCL12 secretion via STAT3 activation. As expected, IFNγ induced the phosphorylation of STAT3 in DFs (Figure 6C). In contrarst, when JAK (Baricitinib) or STAT3 inhibitors (Stattic) were added to the culture, IFNγ-induced CXCL12 production was significantly reduced (Figure 6D, ## *p* < 0.01 vs. control, * *p* < 0.05, ** *p* < 0.01 vs. IFNγ treated). Further investigation on the effect of αCXCL12 in IFNγ-induced AA was conducted using mouse vibrissae organ culture models. IFNγ treatment significantly reduced the hair growth, whereas αCXCL12 treatment increased the length of mouse vibrissae (Figure 6E, # *p* < 0.05 vs. control * *p* < 0.05 vs. IFNγ treated, n = 8). Collectively, these findings suggest that the CXCL12 secreted from DFs plays a role in hair loss caused by INFγ and that αCXCL12 might be useful in AA treatment.

## 3. Discussion

The CXCL12/CXCR4 pathway is known to be involved in chronic skin inflammatory diseases such as psoriasis [18,19]; however, its effects on hair inflammation have not been studied. In a previous study, we investigated the functional role of CXCL12 in hair cycle regulation. CXCL12 levels were elevated during the hair regression period and hair loss was induced by the CXCL12 secreted from DFs. Therefore, we investigated in the present study whether the CXCL12 secreted from DFs near the HFs mediates the inflammation induced by androgens or IFNγ, which is considered a major factor in hair thinning and hair loss. Consequently, we developed a new neutralizing antibody for CXCL12 for commercialization and investigated in the present study whether this neutralizing antibody for CXCL12 was effective in pathological hair loss conditions such as AGA and AA. Androgens such as testosterone and DHT increased CXCL12 in DFs through AR activation and subsequently induced hair miniaturization. In the contrast, αCXCL12 significantly promoted hair growth in AGA mice and increased the hair length in hair follicle organ culture. In an AA model, CXCL12 expression was also significantly increased in DFs. IFNγ treatment significantly increased the CXCL12 level in a concentration-dependent manner via the STAT3 pathway. However, treatment of αCXCL12 improved hair loss in AA and reduced the CD8^+^ cells around HFs. Likewise, αCXCL12 delayed the onset of AA. Collectively, these results indicate that αCXCL12 is promising in the treatment of AGA and AA.

Hair loss can result from various factors, and current treatments such as topical medications, oral drugs, and surgical procedures offer temporary relief but require continuous usage or repeated interventions. Developing a long-lasting solution is crucial to enhance the effectiveness and convenience of hair loss treatments. Researchers are exploring innovative formulations that can provide sustained benefits over an extended period. These formulations may include biocompatible materials, nanoencapsulation, and controlled-release technologies to ensure the gradual release of therapeutic agents directly to HFs [20]. In addition to formulation, antibody therapy has many advantages over long-lasting effects in hair loss treatment. Because the molecular weight of the antibody is very high, it is difficult for it to penetrate the capillary in the scalp and it is slowly absorbed through the lymphatic system [21,22]. Therefore, many s.c. antibody medications are administered at regular intervals, often weekly, biweekly, or monthly. In addition, direct injection of antibody medications into the hair loss areas is possible, leading to superior treatment efficacy while minimizing the systemic side effects. The long duration of action and minimal systemic side effects make CXCL12 antibody therapy highly promising as a hair loss treatment.

AGA, commonly known as male-pattern baldness, is a hair loss condition influenced by genetic and hormonal factors. We have investigated the interaction of CXCL12 with androgens in the pathogenesis of AGA and found that elevated levels of androgen were associated with an increase in CXCL12 expression in DFs. Upon the binding of androgens to the AR, the activated AR translocates to the nucleus, where it may interact with specific DNA sequences known as AREs to activate CXCL12 expression [6]. This increase in CXCL12 levels might influence hair follicle maintenance and contribute to the hair miniaturization observed in AGA. Although we did not further investigate the specific localization of CXCL12, it is reportedly expressed in DFs, in addition to ORS cells, endothelial cells, and T cells surrounding HFs [23]. The CXCL12 secreted by various cells within the follicular microenvironment interacts with its receptor CXCR4, which is located primarily in DPCs. CXCL12/CXCR4 signaling ultimately transits HFs from the anagen (growth) phase to the catagen (regression) phase of the hair cycle, disrupting the normal hair growth process [2].

AR is a pivotal transcription factor primarily associated with the effects of androgen hormones like testosterone and DHT. The AR was highly expressed in the cells of the sebaceous glands and weakly detected or not detected in the epithelial components of HFs. Of interest, the AR is reportedly expressed in DPCs and testosterone and DHT inhibited hair regrowth by activating the AR in DPCs [14]. For example, the nuclear localization of the AR was significantly increased in the DPCs of AGA balding scalps and the AR induced the regression of blood vessels in the DP to mediate hair loss [24]. DPCs from the balding scalps of AGA patients underwent premature senescence, showing a senescent phenotype and high AR expression [25]. However, we first demonstrated a different hair loss mechanism where androgens primarily increased the CXCL12 secretion from DFs, which increased the expression of the AR and CXCR4 in DPCs to induce hair loss.

It has been reported that testosterone and the AR might be involved in reducing inflammation. For example, testosterone suppressed hepatic inflammation by downregulating IL-17, CXCL9, and CXCL10 in an acute cholangitis model [26]. AR activity also suppressed activation and reduced proliferation in CD8 T cells [27]. However, androgens and the AR induced moderate to severe inflammatory responses in the scalps of AGA patients [3,28]. This heightened inflammation can contribute to hair follicle miniaturization and, ultimately, lead to hair thinning and loss in affected individuals. Michel et al. performed a microarray analysis, and inflammatory cytokines such as CXCL12 and CCL18 were found to be increased in AGA-affected scalps [3]. Of interest, AR-KO in DFs caused a considerable increase in the expression of several chemokines, including CXCL1, CXCL2, and CXCL10, while decreasing CXCL12 [29]. We further found that AR knockdown in DFs induced the expression of CXCL9, CXCL10, and CXCL11, while reducing CXCL12 (Appendix A). Since CXCL12 expression is high in AGA patients and induced by the AR, we can hypothesize that AR-mediated CXCL12 could induce an inflammatory response in AGA.

IFNγ is a cytokine that plays a crucial role in immune responses and reportedly has effects on the regulation of CXCL12, a chemokine involved in immune cell trafficking and inflammation. For example, CXCL12 was induced by inflammatory factors such as IFNγ and TNF-α and related to CXCR4^+^ immune cell infiltration and angiogenesis in periodontal tissues [30]. Although we did not investigate the phosphorylation of other STATs, STAT3 phosphorylation reportedly regulates CXCL12 expression. For example, activation of STAT3 mediated CXCL12 upregulation in the dorsal root ganglion [31] and was essential for CXCL12-induced cell invasion in bladder cancer [32]. In the AA condition, it is reasonable to assume that IFNγ binding to its receptor in DFs can trigger the activation of the STAT3 signaling pathway, where it binds to the promoter region of the CXCL12 gene and promotes transcriptional activity, resulting in increased CXCL12 expression. The upregulation of CXCL12 in DFs by IFNγ highlights its role in immune responses in AA, and CXCL12/CXCR4 signaling guides immune cells to the vicinity of HFs. These immune cells highly express CXCR4 and CD4^+^ and CD8^+^ T cells infiltrate the follicular structures and mount an attack on HF epithelial cells. In the present study, αCXCL12 primarily downregulated CD8^+^ T cells in AA skin, which recovered the peripheral immune-privileged status to protect the HF.

Currently, there are a few treatments approved and available for AGA and AA in clinics. Finasteride is a representative drug in this category, working by inhibiting the activity of the 5-α reductase enzyme in the scalp. JAK inhibitors represent a relatively recent approach to hair loss treatment, showing particular effectiveness in AA. These drugs work by suppressing the inflammatory reactions that can cause HF damage. In addition to these two medications, our present study suggests that CXCL12 antibody therapy holds promise for treating both AGA and AA in the clinic.

## 4. Materials and Methods

### 4.1. Cell Isolation and Cell Culture

Individual HFs were generously provided by Dr. Jino Kim (New Hair Institute, Seoul, Republic of Korea). All biopsies were performed with full written consent from the patients. Furthermore, the ethical and scientific committees of the participating institution confirmed that the current study adhered to the ethical standards outlined in the 1964 Declaration of Helsinki.

Human DPCs were microdissected from the HFs under a stereomicroscope. The lower part of the bulb was inverted using forceps and needles, and any remaining epithelium-derived tissue was removed to expose the DP. The DP was then attached to a 35 mm culture dish (CellBIND^®^, Corning, New York, NY, USA) and cultured without disturbance for ten days. DPCs were cultured in DPC growth medium (PromoCell, Heidelberg, Germany) with 1% antibiotic-antimycotic (Thermo Fisher Scientific, Waltham, MA, USA).

Human DFs were cultured in Dulbecco’s modified Eagle’s medium (DMEM; Gibco, Grand Island, NY, USA) with 10% fetal bovine serum (FBS; Hyclone, Logan, UT, USA) and maintained in a humidified incubator at 37 °C with 5% CO_2_.

### 4.2. Preparation of the CXCL12 Neutralizing Antibody

To generate a monoclonal antibody against CXCL12, 50 µg of human recombinant CXCL12 protein (Sino Biological Inc., Beijing, China) was intraperitoneally injected into 6-week-old female Balb/c mice using complete Freund’s adjuvant (Sigma-Aldrich, St. Louis, MO, USA). The subsequent immunization was repeated three times, with the same amount of immunogen emulsified in incomplete Freund’s adjuvant (Sigma-Aldrich) at weekly intervals. A hybridoma was created by fusing immunized mouse splenocytes with SP2/0 cells, a mouse myeloma cell line, and screened in hypoxanthine, aminopterin, and thymidine (HAT) medium containing HAT supplement (Gibco) in DMEM medium. The antibody was purified using Protein A Sepharose and SP Sepharose columns (Invitrogen, Carlsbad, CA, USA). The amino acid sequence encoding the anti-CXCL12 antibody was determined through nucleic acid sequencing. We conducted antibody sequencing for the hybridoma CXCL12 and analyzed the reactivity of four clones. The results showed that three clones were identified as the same antibody, whereas one was determined to be a different antibody with a distinct amino acid sequence. As a result, we successfully cloned two monoclonal antibodies recognizing CXCL12. The clone exhibiting the highest binding affinity with the antigen was selected and utilized in this study.

### 4.3. Reverse Transcription-Quantitative Polymerase Chain Reaction (RT-qPCR) Assay

Quantitative real-time PCR (qRT-PCR) reactions were conducted using the StepOne Real-Time PCR System (Applied Biosystems, Foster City, CA, USA). Total cellular RNA was extracted using Invitrogen TRIzol Reagent (Thermo Fisher Scientific) and then subjected to reverse transcription using a cDNA synthesis kit (Nanohelix, Daejeon, Republic of Korea). The primer sequences used for the experiments were as follows: (forward and reverse), respectively: 5′-ACTACACCGAGGAAATGGGCT-3′ and 5′-CCCACAATGCCAGTTAAGAAGA-3′ for human CXCR4, 5′-CCAGGGACCATGTTTTGCC-3′ and 5′-CGAAGACGACAAGATGGACAA-3′ for human AR, and 5′-ATTCTCAACACTCCAAACTGTGC-3′ and 5′- ACTTTAGCTTCGGGTCAATGC-3′ for human CXCL12.

### 4.4. Western Blotting

For the preparation of whole-cell extracts, adherent DPCs or DFs were washed with PBS, detached by scraping, and lysed in RIPA (Biosesang, Seoul, Republic of Korea). The total protein and nuclear fractions were separated using sodium dodecyl sulfate-polyacrylamide gel electrophoresis (SDS-PAGE) using 10% gels and then transferred to PVDF membranes (Millipore, Billerica, MA, USA). The membranes were blocked with 5% fat-free dried milk in TBS-T (0.1% Tween 20 in Tris-buffered saline) for 1 h at room temperature and subsequently incubated overnight with primary antibodies at 4 °C. The next day, the membranes were washed three times with TBS-T and incubated with HRP-conjugated secondary antibodies for 1 h at room temperature. The membrane was then treated with an enhanced chemiluminescence solution (Millipore) and imaged. The quantification of the protein bands was performed using Image J. The primary antibodies used are listed in Table 1.

### 4.5. Immunofluorescence Staining for Cells

Fixed hDFs were treated with primary antibodies specific for p-STAT3 (1:100) or the AR (1:100) at 4 °C. The next day, the cells were washed with PBS-T and incubated with Alexa Fluor 488 goat anti-rabbit IgG (1:500) for 1 h at room temperature. Nuclei were counterstained with DAPI. Immunofluorescence images were captured using a Nikon Eclipse Ts2 microscope (Nikon, Tokyo, Japan).

### 4.6. siRNA Transfection

For AR knockdown, DF cells were seeded in 60 mm dishes. The following day, 20 μM of control or AR siRNA (Bioneer, Daejeon, Republic of Korea) was transfected using Lipofectamine RNAi MAX (Invitrogen) according to the manufacturer’s instruction. Cells were incubated for 48 h after transfection, and AR knockdown efficiency was evaluated by using qRT-PCR.

### 4.7. CRISPR/Cas9 Transfection

For AR knockout, DFs were seeded in 60 mm dishes. The following day, 1 μg of control or AR CRISPR/Cas9 KO plasmid (Santa Cruz, CA, USA) was transfected using Lipofectamine 2000 (Invitrogen). Cells were incubated for 72 h after transfection, and the AR silencing were evaluated by using Western blotting.

### 4.8. Measurement of CXCL12 via Enzyme-Linked Immunosorbent Assay

DF cells were seeded in 60 mm dishes and treated with varying concentrations of DHT, TP, or IFNγ. After 48 h of culture, the conditioned medium was collected and centrifuged to remove debris. The CXCL12 content was quantified using ELISA kits (R&D Systems, Minneapolis, MN, USA) following the manufacturer’s instructions.

### 4.9. Animals

Male mice aged 4 to 7 weeks and female adult mice aged 9 weeks from the C_3_H/HeN strain were procured from Orient Bio Co. Ltd. (Sungnam, Republic of Korea). All animal studies were carried out with the approval of the Animal Care and Use Committee at the International Medical Center, adhering to ethical guidelines to minimize suffering and ensure the wellbeing of the animals.

### 4.10. Hair Organ Culture

The hair growth activity of mouse vibrissae and adult scalp HFs was observed during organ culture. The HFs were isolated and cultured according to the method previously described by Jindo and Tsuboi [33]. The Institutional Review Board of Epibiotech approved specimen collection (IRB; 70094321–202108-BR-001–01). Informed consent was obtained from all subjects involved in the study. Normal anagen vibrissae HFs and adult scalp HFs were obtained from the upper lip region using a scalpel and forceps. Isolated HFs were placed in a defined medium (Williams E medium supplemented with 2 mM L-glutamine, 10 µg/mL insulin, 10 ng/mL hydrocortisone, 100 U/mL penicillin, and 100 µg/mL streptomycin, without serum). Individual vibrissae HFs and adult scalp HFs were photographed 48–120 h after the start of the incubation. Changes in hair length were calculated from the photographs and expressed as the mean ± SE of 8–10 vibrissae HFs or human scalp HFs.

### 4.11. Androgenic Alopecia Animal Model

Animal experiments were performed according to Kim [34] and Paus et al. [35], pharmacopoeia and the Institutional Animal Care and Use Committee of Yonsei University (IACUC-202302-1636-01). For the TP-induced AGA model, the dorsal area (2.5 cm × 4 cm) of 7-week-old C_3_H/HeN male mice (telogen stage of the hair cycle) was shaved using a clipper and an electric shaver, with special care taken to avoid damaging the skin. From the next day, mice were subcutaneously administered TP (0.5 mg/day, dissolved in corn oil; Sigma) or corn oil alone (control group) five times per week. αCXCL12 (5–20 μg/head) was administered twice per week subcutaneously. A mixed solvent of propylene glycol:ethanol:water in a ratio of 3:5:2 was employed as the vehicle. Topical treatment with 0.1 mL of 0.05% dutasteride in the mixed solvent was administered daily, serving as the positive control. Any darkening of the skin was carefully monitored via photography. After 3 weeks, the dorsal hair was shaved and weighed.

For the DHT-induced AGA model, a single dose of DHT (5 mg per head, dissolved in corn oil; Sigma) or corn oil alone (control group) was subcutaneously administered to the dorsal area of C_3_H/HeN male mice in the telogen stage of the hair cycle. Four days later, all the mice were shaved using a clipper and an electric shaver, with special care taken to avoid damaging the skin. αCXCL12 (20 μg/head) was subcutaneously administered twice per week. Any darkening of the skin was carefully monitored via photography. After 2 weeks, the dorsal hair was shaved and weighed.

### 4.12. Alopecia Areata Animal Model

The AA mice model was induced by transferring in vitro-expanded SDLN cells from AA-affected mice as previously described [36,37]. In brief, SDLN cells were extracted from mice that had spontaneously developed AA and cultured in advanced RPMI 1640 (Gibco) supplemented with 10% FBS (Hyclone), 2 mM GlutaMAX (Gibco), and 100 U/mL penicillin–streptomycin (Hyclone). The culture medium was further supplemented with IL-2 (Roche, Basel, Switzerland), IL-7 (R&D Systems), and IL-15 (R&D Systems). The cells were stimulated using Dynabeads Mouse T-Activator CD3/CD28 (Thermo Fisher Scientific) and intradermally transferred to at least 10-week-old C_3_H/HeN female mice with a normal hair coat during the second telogen phase.

### 4.13. Immunofluorescence Staining for Paraffin Sections

Mouse skin tissue slides were de-paraffinized and subjected to antigen retrieval by incubation in boiling antigen retrieval solution (pH 6.0; Dako, CA, USA) using a microwave oven for 10 min. The slides were then washed with 0.05% Triton in phosphate-buffered saline (PBS-T) and incubated overnight at 4 °C with the following primary antibodies: anti-CXCL12 (1:100, R&D systems), anti-p-STAT3 (1:100), anti-CD8 (1:100), anti-MHC class I (1:100), and anti-MHC class II (1:100). Subsequently, the slides were incubated with secondary antibodies, including Alexa Fluor 488 goat anti-mouse IgG (1:500; Invitrogen), Alexa Fluor 488 goat anti-rabbit IgG (1:500; Invitrogen), Alexa Fluor 594 goat anti-rabbit IgG (1:200; Invitrogen), or Alexa Fluor 594 goat anti-mouse IgG (1:200, Invitrogen), for 1 h at room temperature along with 4,6-diamidino-2-phenylindole (DAPI; Sigma-Aldrich). Immunofluorescence images were acquired using a Nikon Eclipse Ts2 microscope.

### 4.14. Flow Cytometry Analysis

SDLN cells were isolated by grinding through a 40 μm strainer to achieve single cell isolation. CD8^+^ T cells were separated from SDLN cells using anti-PE microbeads (Miltenyi Biotec, Bergisch Gladbach, Germany). For live/dead staining, the Zombie NIR fixable viability kit (BioLegend, San Diego, CA, USA) was used for 10 min. To block nonspecific binding, cells were incubated with a CD16/CD32 antibody (BD Biosciences, San Jose, CA, USA) for 10 min. For surface staining, the cells were labeled with the following antibodies: PE anti-CD8, anti-CD4; FITC anti-CD8, anti-CD3; APC anti-NKG2D; PerCP-Cy5.5 anti-CD45 (all purchased from BioLegend). The samples were acquired using CytoFLEX (Beckman Coulter, Miami, FL, USA), and the data were analyzed with CytExpert software (version 2.4.0.28).

### 4.15. Statistical Analysis

All data are presented as the mean ± standard deviation based on three independent experiments. Student’s *t*-test was used when comparing between two groups and one-way ANOVA with Tukey’s post hoc test was used for comparing multiple groups. A *p*-value of less than 0.05 was considered statistically significant. All statistical analyses were carried out using GraphPad Prism (Version 5.01, La Jolla, CA, USA).

## Figures and Tables

**Figure 1 ijms-25-01705-f001:**
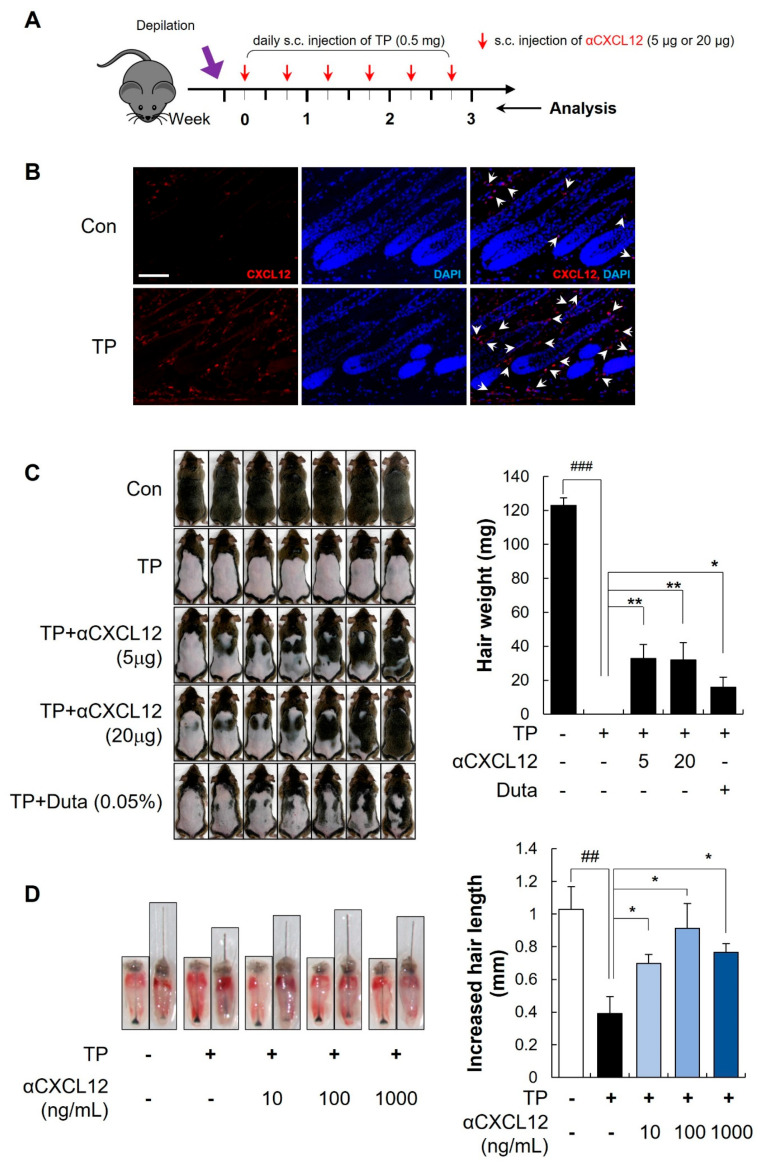
Neutralization of CXCL12 promotes hair growth in a testosterone-induced AGA model. (**A**) The back skin of 7-week-old C_3_H male mice was shaved and subcutaneous (s.c.) injections of testosterone propionate (TP) were used to induce the AGA. (**B**) The expression of CXCL12 in the dorsal skin of control and TP-treated mice was detected using immunofluorescence. CXCL12^+^ cells (red) are indicated by white arrows, and DAPI staining (blue) indicates cell nuclei. The scale bar is set at 100 μm. (**C**) Different doses of CXCL12 neutralizing antibody (αCXCL12, 5, and 20 μg) were subcutaneously injected twice a week for three weeks. A daily topical treatment of 0.05% dutasteride served as a positive control. αCXCL12 administration accelerated hair growth in TP-treated C_3_H mice and the hair weight was measured. ### *p* < 0.001 vs. control, * *p* < 0.05, ** *p* < 0.01 vs. TP-treated, n = 7. (**D**) αCXCL12 (10, 100, or 1000 ng/mL) treatment increased the length of mouse vibrissa follicles in the AGA mimic ex vivo model. ## *p* < 0.01 vs. control, * *p* < 0.05 vs. TP-treated, n = 8. + or -: treated with or without TP, αCXCL12. The asterisk and sharp symbols indicate statistical differences using Student’s *t*-test.

**Figure 2 ijms-25-01705-f002:**
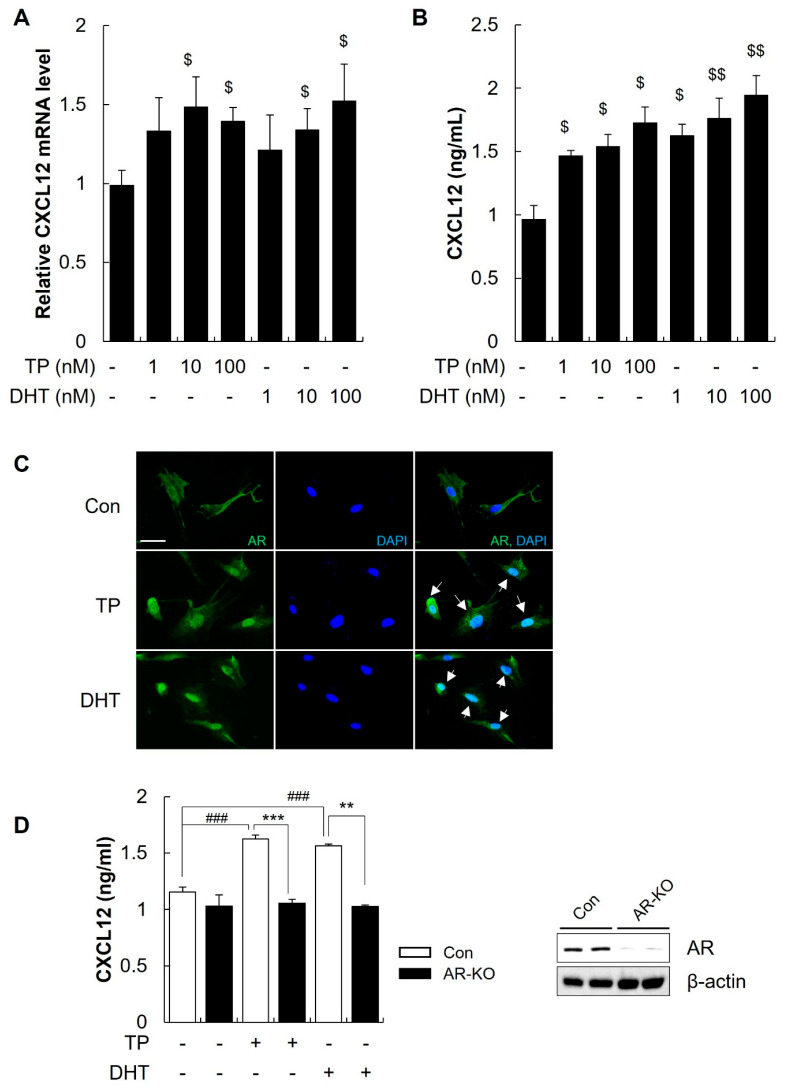
Androgen treatment enhances CXCL12 expression in dermal fibroblasts. (**A**) Dermal fibroblasts (DFs) were treated with various concentrations of TP (1, 10, and 100 nM) and DHT (1, 10, and 100 nM) for 24 h, and the expression of CXCL12 was assessed using qRT-PCR. $ *p* < 0.05 vs. control. (**B**) After treating DFs with different concentrations of TP and DHT for 48 h, the culture medium was collected and the secreted CXCL12 levels were quantified using ELISA. $ *p* < 0.05, $$ *p* < 0.01 vs. control. The dollar sign ($) indicates differences in one-way ANOVA. (**C**) Immunostaining revealed that the translocation of the AR (green) in DFs increased after TP (100 nM) and DHT (100 nM) treatment for 1 h, as indicated by the white arrows. DAPI staining (blue) marks the cell nuclei. The scale bar is set at 50 μm. (**D**) After AR-CRISPR/Cas9 knockout (AR-KO) for 48 h, DFs were treated with TP and DHT for an additional 48 h to collect the culture medium for ELISA analysis. AR-KO significantly reduced the secretion of CXCL12 from DFs. Western blot analysis indicated differences in AR expression between the control and AR-KO groups. ### *p* < 0.001 vs. control, ** *p* < 0.01, *** *p* < 0.001 vs. TP or DHT treatment. The asterisk and sharp symbols indicate statistical differences using Student’s *t*-test.

**Figure 3 ijms-25-01705-f003:**
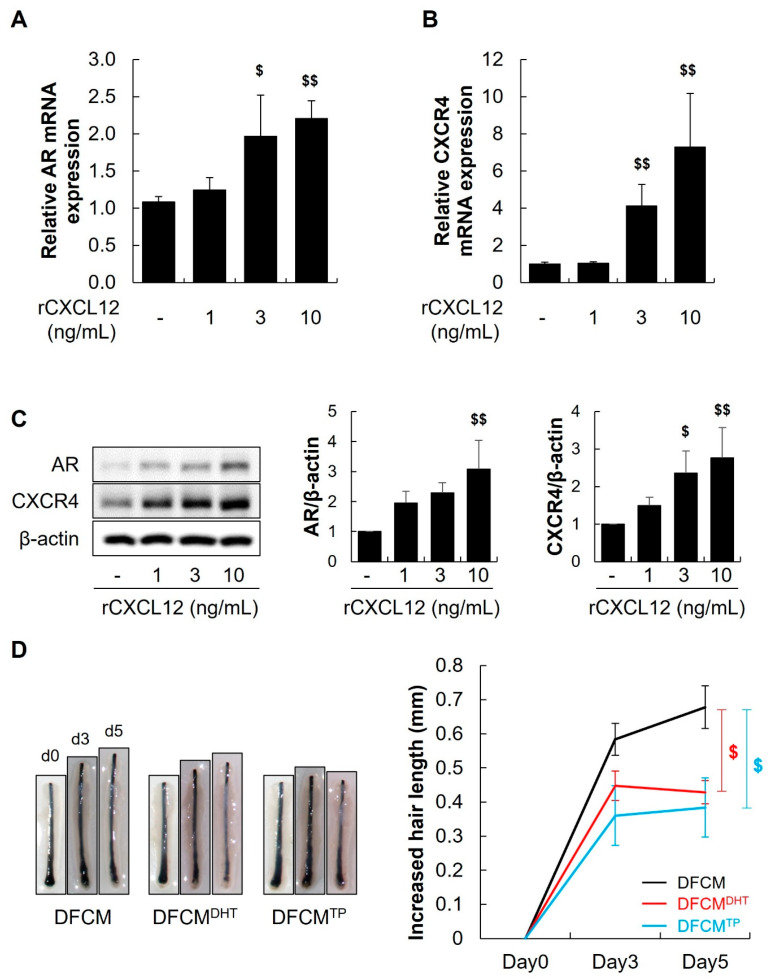
CXCL12 secreted from DFs induces AR and CXCR4 in DPCs. The effects of rCXCL12 on the expression of AR and CXCR4 in DPCs were observed using qRT-PCR (**A**,**B**) and Western blot analysis (**C**). rCXCL12 increased the mRNA and protein expression of the AR and CXCR4 in DPCs. $ *p* < 0.05, $$ *p* < 0.01 vs. Control. (**D**) DFs were treated with 100 nM TP or DHT for 48 h and the culture medium (CM) was collected. CM from DFs treated with TP and DHT (DFCM^TP^ and DFCM^DHT^) significantly reduced hair length in human hair organ culture. $ *p* < 0.05 vs. DFCM, n = 10. The dollar sign ($) indicates differences in a one-way ANOVA.

**Figure 4 ijms-25-01705-f004:**
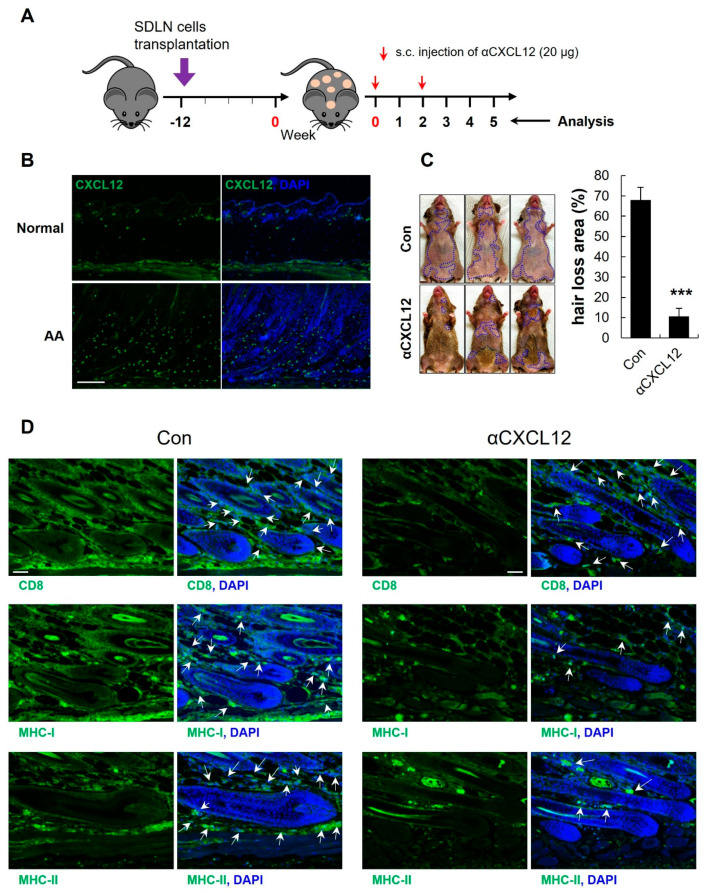
CXCL12 neutralization improves alopecia areata. (**A**) In this experimental design, skin-draining lymph node (SDLN) cells were isolated from AA-affected C_3_H/HeN female mice and intradermally injected into the dorsal skin of naïve mice to induce AA. Severe hair loss was observed after 12 weeks, and αCXCL12 (20 μg) was subcutaneously injected twice in a 2-week interval. (**B**) Skin sections from normal and AA mice were stained with an anti-CXCL12 antibody (green). The expression of CXCL12 increased in the AA model. The scale bar is set at 500 μm. (**C**) The administration of αCXCL12 in AA mice significantly reduced the areas of hair loss, which are marked in the images with dotted lines. The extent of hair loss in both the control and αCXCL12-treated mice was quantified using Image J (v1.53t). *** *p* < 0.001 vs. control, n = 3. (**D**) Skin sections of AA and αCXCL12-treated mice were stained with anti-CD8, anti-MHC-I, or anti-MHC-II antibodies. The αCXCL12-treated group had reduced expression of these immune reaction markers (green; white arrows). DAPI staining (blue) indicates cell nuclei. The scale bar is set at 100 μm. An asterisk indicates a statistical difference using Student’s *t*-test.

**Figure 5 ijms-25-01705-f005:**
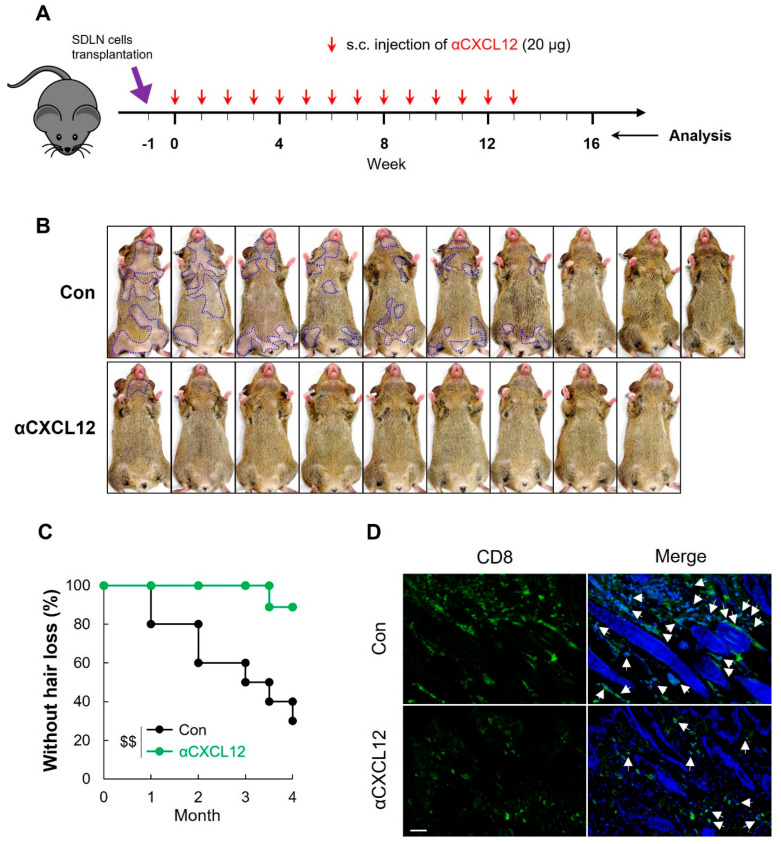
CXCL12 neutralization prevents alopecia areata onset. (**A**) In this experimental design, 10-week-old C_3_H female mice were treated with αCXCL12 (20 μg) through subcutaneous injections once a week for 14 weeks following AA-affected SDLN cell transplantation. (**B**) αCXCL12 treatment significantly delayed the onset of AA, which was marked with dotted lines. (**C**) The incidence of AA onset in SDLN-transplanted C_3_H mice. Control (n = 10) or αCXCL12 treated (n = 9). (**D**) Skin sections were stained with an anti-CD8 antibody, and αCXCL12 reduced the expression of CD8^+^ cells (green; white arrows). DAPI staining (blue) indicates the cell nuclei. The scale bar is set at 100 μm. $$ *p* < 0.01 vs. control. The dollar sign ($) indicates differences in one-way ANOVA.

**Figure 6 ijms-25-01705-f006:**
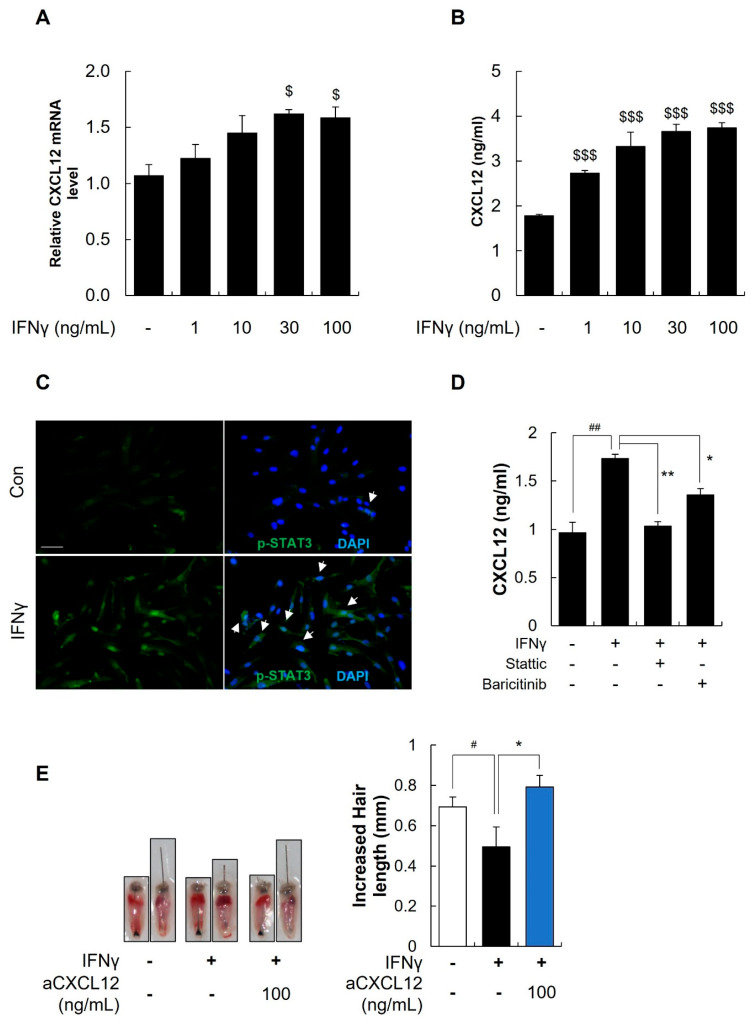
Involvement of JAK/STAT signaling in IFNγ-mediated CXCL12 secretion. (**A**) Dermal fibroblasts (DFs) were treated with IFNγ at various concentrations (1, 10, 30, and 100 ng/mL), and the expression of CXCL12 was significantly increased. $ *p* < 0.05 vs. control. (**B**) The secreted CXCL12 level was measured using ELISA, and IFNγ significantly increased CXCL12 secretion. $$$ *p* < 0.001 vs. control. The dollar sign ($) indicates differences in one-way ANOVA. (**C**) Immunostaining showed that p-STAT3 (green) in DFs was increased by IFNγ (100 ng/mL), as indicated by the white arrows. DAPI staining (blue) indicates the cell nuclei. The scale bar is set at 100 μm. (**D**) JAK inhibitor (Baricitinib) and STAT3 inhibitor (Stattic) treatment attenuated the IFNγ-induced CXCL12 secretion from DFs. ## *p* < 0.01 vs. control, * *p* < 0.05, ** *p* < 0.01 vs. IFNγ treated. (**E**) αCXCL12 treatment increased the length of mouse vibrissae follicles in the AA mimic ex vivo model. # *p* < 0.05 vs. control * *p* < 0.05 vs. IFNγ treated, n = 8. Asterisk and sharp indicate statistical differences using Student’s *t*-test.

**Table 1 ijms-25-01705-t001:** Primary antibodies used for immunofluorescence analysis and western blot.

Antibodies	Source	Dilution (IF)	Dilution (WB)	Used to Identify
p-STAT3^Tyr705^	Cell signaling; #9145T	1:100		DF cell
AR	Santa cruz; sc-7305	1:100	1:1000	DP, DF cell
CXCR4	NOVUS; NB100-56437		1:1000	DP cell
CD8	Santa cruz; sc-1177	1:100		Mouse skin tissue
MHC-I	Santa cruz; sc-55582	1:100		Mouse skin tissue
MHC-II	Santa cruz; sc-32247	1:100		Mouse skin tissue
CXCL12	R&D systems; MAB350-SP	1:100		Mouse skin tissue

## Data Availability

The original contributions presented in the study are included in the article/Appendix A, further inquiries can be directed to the corresponding authors.

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
