# Peer review of "CXCL12 Neutralizing Antibody Promotes Hair Growth in Androgenic Alopecia and Alopecia Areata"

_ijms, 2024, doi:10.3390/ijms25031705_

Round 1
Reviewer 1 Report
Comments and Suggestions for Authors
I have reviewed the paper by Zheng et al.
The results are solid, and the findings are impressive.
Nevertheless, it is not clear how they relied on a single monoclonal antibody. Where other clones created and evaluated? How was it determined that this αCXCL12 was neutralizing? How was the dosage of 5 and 20ug established? How did they come up with the twice per week subcutaneous protocol?
Why the study period is 4months in the Kaplan-Meier curves, with no change in the first month, when the ex vivo model showed hair length increasing in just a few days.
A pathway model, along with the Discussion would be a good idea to consider.
Reviewer 2 Report
Comments and Suggestions for Authors
Based on their previous research on the impact of CXCL12 on the hair cycle, the authors examine the potential therapeutic implications of their previous results - they assess the impact of blocking the cytokine with a dedicated monoclonal antibody. They used in vitro methods as well as experimental models of androgenetic alopecia and alopecia areata.
The work is very interesting. However, there are some minor issues that need clarification:
- the results section lacks numerical data with statistical significance values, only references to figures.
- Figure 1 does not indicate that there was a statistically significant difference between the tested antibody and topical dutasteride
- the methodology does not provide information on the use of dutasteride - substance, vehicle, frequency of application, etc.
Round 2
Reviewer 1 Report
Comments and Suggestions for Authors
All my comments have been answered in an acceptable form.
Paper is acceptable for publications.